# Biotransformation of Ganoderic Acid A to 3-*O*-Acetyl Ganoderic Acid A by Soil-isolated *Streptomyces* sp.

**Te-Sheng Chang** [1,*] , **Horng-Huey Ko** [2] , **Tzi-Yuan Wang** [3] , **Chun-Hsien Lee** [1] **and Jiumn-Yih Wu** [4,*]

1   Department of Biological Sciences and Technology, National University of Tainan, No. 33, Sec. 2, Shu-Lin St., Tainan 70005, Taiwan; derekww2719@gmail.com
2   Department of Fragrance and Cosmetic Science, College of Pharmacy, Kaohsiung Medical University, No. 100, Shih-Chuan 1st Road, Kaohsiung 80708, Taiwan; hhko@kmu.edu.tw
3   Biodiversity Research Center, Academia Sinica, Taipei 115, Taiwan; tziyuan@gmail.com
4   Department of Food Science, National Quemoy University, No. 1, University Road., Jin-Ning Township, Kinmen County 892, Taiwan
*   Correspondence: mozyme2001@gmail.com (T.-S.C.); wujy@nqu.edu.tw (J.-Y.W.)
    Tel.: +886-6-2602137 (T.-S.C.); +886-82-313310 (J.-Y.W.); Fax: +886-6-2602137 (T.-S.C.); +886-82-313797 (J.-Y.W.)

**Abstract:** The medicinal fungus *Ganoderma lucidum* contains many bioactive triterpenoids, ganoderic acid A (GAA) being one of the major ones. The present study explored the microbial biotransformation of GAA, isolating 283 strains of soil actinomycetes and determining their abilities to biotransform GAA with ultra-performance liquid chromatography analysis. One positive strain, AI 045, was selected to validate the biotransformation activity. The strain was identified as *Streptomyces* sp. based on the sequenced 16S rRNA gene. The produced compound obtained from the biotransformation of GAA was purified with the preparative high-performance liquid chromatography method and identified as 3-*O*-acetyl GAA based on mass and nuclear magnetic resonance spectral data. The present study is the first report that bacteria have the novel ability to biotransform the triterpenoids of fungus *G. lucidum*. Moreover, the identified 3-*O*-acetyl GAA is a new triterpenoid product discovered in microbes.

**Keywords:** actinomycetes; biotransformation; *Ganoderma lucidum*; ganoderic acid A; *Streptomyces*

## 1. Introduction

*Ganoderma lucidum* (in Chinese, "Lingzhi") has been used as a nutritional supplement in immunomodulatory and antitumor activities [1]. The medicinal fungus contains many bioactive constituents, such as polysaccharides and triterpenoids. To date, more than 300 different triterpenoids have been isolated and identified from *Ganoderma* spp., of which ganoderic acid A (GAA) was the first identified triterpenoid [2,3]. *Ganoderma lucidum* has been obtained from submerged culture. However, the quantity of *Ganoderma* triterpenoids obtained from the fermentation process varied dramatically based on fermentation parameters, such as pH and dissolved oxygen [4]. Due to a unique component in *Ganoderma*, GAA is used as a marker component for evaluating *Ganoderma* quality. Some bioactivities of GAA have been proven [5–10]. In particular, GAA was proven to play a role in antitumor activity in highly invasive human breast cancer cells [5], human liver cancer cells [6], and human hepatocellular carcinoma [7]. GAA was found to suppress the growth and invasiveness of human breast cancer cells by inhibiting transcription factor activator protein-1 (AP-1) and nuclear factor-κB (NF-κB), resulting in the down-regulation of cyclin-dependent kinase 4 (Cdk4) expression and the suppression of the secretion of urokinase-type plasminogen activator (uPA), respectively [5].

Searching new compounds is an important step in the development of new drugs. In addition to synthetic and natural sources, microbial biotransformation is an alternative way to obtain new compounds. Precursor compounds were biotransformed by microorganisms or enzymes to form new compounds. Additionally, some microbial transformations can modify the functional groups of precursors to improve the bioactivities of the precursor compounds [11–14]. For example, soy isoflavone daidzein was hydroxylated by *Aspergillus oryzae* to form 6-hyroxydaidzein or 8-hydroxydaidzein. These two compounds have not been identified in nature. Moreover, the two compounds possessed over 10-fold more anti-tyrosinase activity than their precursor daidzein [15]. As described above, many *Ganoderma* triterpenoids have been isolated and studied; however, few studies have been published on the microbial biotransformation of *Ganoderma* triterpenoids. Thus, we are interested in investigating the biotransformation of *Ganoderma* triterpenoids. We recently isolated an intestinal bacterium from zebrafish and found that the strain could biotransform GAA to GAA-15-*O*-β-glucoside [16]. The present study isolated 283 soil actinomycetes and determined their ability to catalyze GAA. One of the strains was selected for further identification with an in vitro functional assay. The new biotransformation product was purified with preparative high-performance liquid chromatography (HPLC) and identified via spectra methods.

## 2. Results and Discussion

To analyze their ability to biotransform *Ganoderma* triterpenoid GAA, soil actinomycetes were isolated on AI agar and then cultivated in AI broth with GAA. The cultivation broth was analyzed using UPLC. A total of 283 strains were screened, and one strain (AI 045) was selected for functional studies.

The AI 045 strain was identified through the partial 16S rRNA gene, which was amplified and sequenced by polymerase chain reaction (PCR) with the bacteria-specific 27F (5′-AGAGTTTGATCCTGGCTCAG-3′) and 1391R (5′-GACGGGCRGTGWGTRCA-3′) primer set. The DNA sequence is deposited in NCBI under accession number MH756176. The partial 16S rRNA gene sequences were then blasted against NCBI non-redundant nucleotides to identify the strain. The phylogenetic tree from the blasted results indicated that the AI 045 strain was classified as *Streptomyces* sp. (Figure 1).

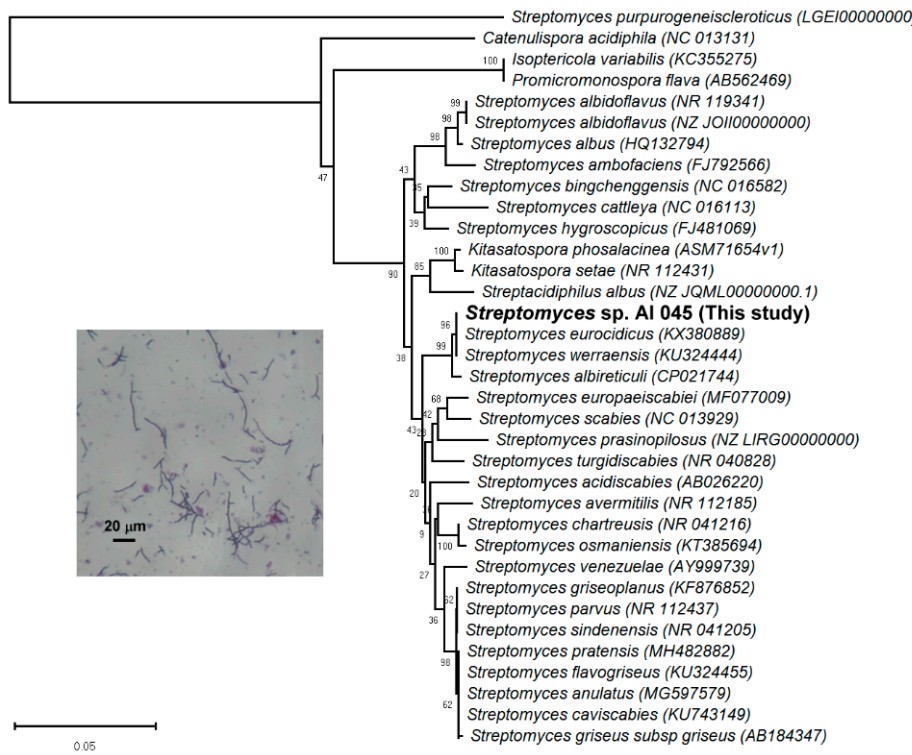

**Figure 1.** Molecular phylogenetic analysis of strain AI 045 and the other candidates with the neighbor-joining (NJ) method based on the general time reversible (GTR+G+I) model [17]. The optimal tree with the sum of branch length = 0.67102039 is shown. The percentage of replicate trees in which the associated taxa clustered together in the bootstrap test (500 replicates) are shown next to the branches [18]. The tree is drawn to scale, with the branch lengths in the same units as those of the evolutionary distances used to infer the phylogenetic tree. The evolutionary distances were computed using the maximum composite likelihood method [19], and are in the units of number of base substitutions per site. The rate variation among the sites was modeled with a gamma distribution (shape parameter = 0.76). The differences in the composition bias among sequences were considered in evolutionary comparisons [20]. The analysis involved 35 nucleotide sequences. All positions with less than 95% site coverage were eliminated. That is, no fewer than 5% alignment gaps, missing data, and ambiguous bases were allowed at any position. There were a total of 875 positions in the final dataset. Evolutionary analyses were conducted in Molecular Evolutionary Genetics Analysis (MEGA) X [21]. Gram staining revealed that strain AI 045 is a long, straight filament aerial hyphae, Gram-positive bacterium (see the inserted photo).

To resolve the chemical structure of the biotransformation product, compound (**1**), the biotransformation was scaled up and the product was purified with preparative HPLC. From compound (**1**), 2.9 mg was isolated from a 2.1-L fermentation broth containing 42 mg GAA. The chemical structure of compound (**1**) was identified and characterized using mass, 1D, and 2D NMR spectrum analyses, including DEPT, HSQC, HMBC, NOESY, and COSY. The molecular formula $C_{32}H_{48}O_8$ of compound (**1**) was deduced from HR-ESI-MS ($m/z$ $C_{32}H_{48}O_8$ [M+H] +). The major resonances in the 1H and 13C NMR spectra were similar to those of ganoderic acid C2 (GAC2) [22], except for an acetoxy group at position C-3 instead of a hydroxyl group of GAC2. The 1H- and 13C-NMR signal assignments were conducted accordingly (shown in Figures S1–S4). The key HMBC correlations of compound (**1**) are shown in Figure S5 and the spectroscopic data are listed in Table S1. Based on these spectral data and compared to 1H-NMR and 13C-NMR data, compound (**1**) was characterized as 3-*O*-acetyl GAA (Figure 2).

**Figure 2.** The chemical structure of the biotransformed metabolite compound (**1**), 3-*O*-acetyl ganoderic acid A.

GAA has a 3-oxo functional group, which rarely reacts to the 3-*O*-acetyl group in one step. Cao et al. reported a similar metabolic pathway in 3-*O*-sulfate GAA, which contained a two-stepwise reaction in rats [23]. The first step is reduction of the 3-oxo group of GAA to a 3-hydroxyl group, which produces an intermediate product, GAC2. The second step is *O*-sulfation of the produced GAC2 to form 3-*O*-sulfate GAA. Therefore, the possible reactions to produce 3-*O*-acetyl GAA by the *Streptomyces* sp. AI045 strain might also involve a two-stepwise reaction due to similar side-chain structures. The first step is the same reduction of the 3-oxo group from GAA to form GAC2, but the second step is an *O*-acetylation of the produced GAC2 to form 3-*O*-acetyl GAA. Figure 3 summarized our predicted biotransformation process of GAA by the *Streptomyces* sp. AI 045 strain. We failed to identify the intermediate product, GAC2. The possible reason is that the two reactions described above have different reaction rates in vivo and the first reaction is the rate-determining step and slower than the secondary *O*-acetylation. Thus, once the GAA is biotransformed to GAC2, GAC2 is immediately biotransformed to 3-*O*-acetyl GAA. In this situation, the intermediate GAC2 was not detected during the fermentation process. However, the reality of the proposed 3-*O*-acetylation mechanism needs to be studied in the future.

Since triterpenoids have many bioactivities, many researchers have focused on examining triterpenoids biotransformations [11–14]. Interestingly, this was the first study to find 3-*O*-acetylation in the microbial biotransformation of triterpenoids. Therefore, the enzyme responsible, possibly *O*-acetyltransferase, has not been identified yet. Further study is needed to identify the intact gene from the *Streptomyces* genome.

**Figure 3.** Proposed biotransformation process of ganoderic acid A by the *Streptomyces* sp. AI 045 strain.

## 3. Materials and Methods

### 3.1. Microorganisms and Chemicals

GAA was purchased from Baoji Herbest Bio-Tech (Xi-An, Shaanxi, China). Actinomycete isolation (AI) medium was purchased from Sigma (St. Louis, MO, USA). The composition of the AI medium

is L-asparagine (0.1 g/L), dipotassium phosphate (0.5 g/L), ferrous sulfate (0.001 g/L), magnesium sulfate (0.1 g/L), sodium caseinate (2 g/L) and sodium propionate (4 g/L). The other chemicals, materials needed for PCR, restriction enzymes, reagents and solvents used were of high quality and were purchased from commercially available sources.

### 3.2. Screening and Identifying Soil Bacteria with Biotransformation Activity

Soil actinomycetes were isolated by the method in our previous study using AI agar [24]. Soil samples were plated on AI agar. After cultivation, colonies formed on the plates were transferred to a deep 48-well microplate containing 1 mL of AI medium for a primary culture. A secondary culture was conducted with 10% inoculation of primary culture in the presence of 100 mg/L of antcin K. The secondary cultivation was carried out at 180 rpm and 30 °C for 3 days. An equal volume of ethanol was then added to each well of the secondary cultivation microplate and shaken vigorously for 30 min at 28 °C. The cell debris was removed by centrifugation at 4800 rpm for 30 min. The supernatant from the extracted broth was assayed by ultra-performance liquid chromatography (UPLC) to measure the biotransformation activity.

### 3.3. UPLC Analysis

The UPLC system (Acquity UPLC H-Class, Waters, Milford, MA, USA) was equipped with an analytic C18 reversed-phase column (Kinetex® C18, 1.7 μm, 2.1 i.d. × 100 mm, Phenomenex Inc., Torrance, CA, USA). The operation conditions for UPLC analysis were from our previous study [24].

### 3.4. Candidate Strain Classification Via 16S rRNA Gene Analysis

AI 045 strain was classified using the genetic analysis according to our previous study [24].

### 3.5. Scaled-Up Fermentation, Isolation, and Identification of the Biotransformation Products

Four of the 283 studied strains (AI 034, AI 045, AI 074, and AI 125) were screened and might have similar biotransformation activity in the deep 48-well microplate according to the UPLC analyses (data not shown). One representative *Streptomyces* strain (AI 045) was selected for scaled-up fermentation to isolate the compound from the biotransformation. The AI 045 strain was cultured in a 1000-mL baffled Erlenmeyer flask containing 100 mL of AI medium and 20 mg/L of GAA at 180 rpm, 28 °C for 72 h. A total of 21 flasks of AI 045 cultivation (2.1 L) were conducted. At the end of the fermentation, the broth was extracted with ethyl acetate twice and the ethyl acetate fractions were condensed under a vacuum. The residue, which was resuspended in 50% methanol, was applied to a preparative YoungLin HPLC system (YL9100, YL Instrument, Gyeonggi-do, South Korea), which was operated according to our previous study [24]. Finally, 2.9 mg of the metabolite was obtained and resolved in MeOH-d4. The structure of the compound was confirmed with nuclear magnetic resonance (NMR) and mass spectral analysis. The mass analysis was performed on a Finnigan LCQ Duo mass spectrometer (Thermo Quest Corp., San Jose, CA, USA) with electrospray ionization (ESI). 1H- and 13C-NMR, distortionless enhancement by polarization transfer (DEPT), heteronuclear single quantum coherence (HSQC), heteronuclear multiple bond connectivity (HMBC), nuclear Overhauser effect spectroscopy (NOESY), and correlation spectroscopy (COSY) spectra were recorded on a Bruker AV-700 NMR spectrometer (Bruker Corp., Billerica, MA, USA) at ambient temperature. Standard pulse sequences and parameters were used for the NMR experiments and all chemical shifts were reported in parts per million (ppm, δ).

## 4. Conclusions

Although numerous triterpenoids have been identified from the medicinal fungus *G. lucidum*, few studies have been published on the microbial biotransformation of *Ganoderma* triterpenoids. The present study investigated biotransformation of *Ganoderma* triterpenoid GAA and identified

one soil-isolated *Streptomyces* sp. AI 045, making it the first case in which a microbe was found to biotransform triterpenoid to its 3-*O*-acetyl derivative. This is the first time that the acetylation metabolite 3-*O*-acetyl GAA has been identified.

**Supplementary Materials:** Supplementary materials can be found at http://www.mdpi.com/2311-5637/4/4/101/s1.

**Author Contributions:** Conceptualization, T.-S.C.; data curation, H.-H.K., T.-Y.W., C.-H.L., J.-Y.W., and T.-S.C.; methodology, H.-H.K., T.-Y.W., C.-H.L., J.-Y.W., and T.-S.C.; project administration, T.-S.C.; writing—original draft, H.-H.K., T.-Y.W., J.-Y.W., and T.-S.C.; writing—reviewing and editing, H.-H.K., T.-Y.W., J.-Y.W., and T.-S.C.

**Funding:** This research was financially supported by grants from the National Scientific Council of Taiwan (Project No. MOST 107-2221-E-024-005-).

**Conflicts of Interest:** The authors declare no conflict of interest.

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
