# Peer review of "Biotransformation of Ganoderic Acid A to 3-O-Acetyl Ganoderic Acid A by Soil-isolated Streptomyces sp."

_fermentation, doi:10.3390/fermentation4040101_

Reviewer 1 Report

Abstract

Line 18: "the ability to catalyze GAA..." at this point you have to add the term biotransformation.

Line 19: I think that "functional studies" it is not the most appropriate term, please change it or clarify the meaning.

Introduction

Line 30: "Ganoderma lucidum (in Chinese, “Lingzhi”) has been used as a nutrition supplement in the prevention of some diseases [1]."  

Please specify what kind of disease.

Line 46-47: "no study on the microbial biotransformation of Ganoderma triterpenoids has been reported." 

I found  your work in the literature concerning a biotransformation of Ganoderma triterpenoid (Molecules, 2018, 23, 2345) please change the sentence or include this biblioghaphy.

Line 50: Why did you only choose a strain and don't show the results obtained with the others?

Materials and methods

Line 56: Which is the composition of AI medium?

Line 60-61: Please could you insert some details about the isolation method.

Line 72: "The AI 045 strain was cultured in a 1000-mL baffled Erlenmeyer flask containing 100 mL of AI 72 medium and 20 mg/L of GAA" 

Why did you use such a low substrate concentration?

Line 80: "spectroscopy (NOESY), and correlation spectroscopy (COSY) 83 spectra were recorded on a Bruker AV-700 NMR spectrometer (Bruker Corp., Billerica, MA, USA) at 84 ambient temperature. Standard pulse sequences and parameters were used for the NMR experiments, 85 and all chemical shifts were reported in parts per million (ppm, δ)." 

Which solvent did you use for NMR analysis?

Results and discussion

Line 115-116: "From a 116 2.1-L fermentation broth containing 42 mg GAA, 2.9 mg of compound (1) was isolated." 

This yield is very low and I think that this process should be improved to increase yields. It may be useful to monitor fermentation parameters such as pH, pO2 and temperature and to evaluate the initial substrate concentration.

Line 132-133: "Therefore, the possible reactions to produce 3-O-acetyl GAA by the Streptomyces sp. AI045 strain might be also involved in two-stepwise reaction due to similar side-chain structures." 

It would be very interesting to be able to verify this hypothesis, if it is not possible to recover the reaction intermediate, it would be useful to try to isolate the enzyme.

Author Response

To Reviewer 1:

1.        Line 18: "the ability to catalyze GAA..." at this point you have to add the term biotransformation.

Author’s Reply:

Thank you very much for your comment. We changed the description "the ability to catalyze GAA..." to "their ability to biotransform GAA..." at Line 18 of Page 1 in the revised manuscript.

2.        Line 19: I think that "functional studies" it is not the most appropriate term, please change it or clarify the meaning.

Author’s Reply:

Thank you very much for your comment. We changed the description "for functional studies" to "to validate the biotransformation activity" at Line 20 of Page 1 in the revised manuscript.

3.        Line 30: "Ganoderma lucidum (in Chinese, “Lingzhi”) has been used as a nutrition supplement in the prevention of some diseases [1]."  Please specify what kind of disease.

Author’s Reply:

        Thank you very much for your comment. We changed the sentence to " Ganoderma lucidum (in Chinese, “Lingzhi”) has been used as a nutritional supplement in immunomodulatory and antitumor activities [1]." at Line 32-33 of Page 1 in the revised manuscript.

4.        Line 46-47: "no study on the microbial biotransformation of Ganoderma triterpenoids has been reported."  I found your work in the literature concerning a biotransformation of Ganoderma triterpenoid (Molecules, 2018, 23, 2345) please change the sentence or include this bibliography.

Author’s Reply:

        Thank you very much for your comment. We added our resent discovery as the sentence “We recently isolated an intestinal bacterium from zebrafish and found that the strain could biotransform GAA to GAA-15-O-b-glucoside [16]. ” at Line 64-66 of Page 2 and also added the reference in the revised manuscript.

5.        Line 50: Why did you only choose a strain and don't show the results obtained with the others?

Author’s Reply:

        Thank you very much for your comment. During the screening of soil actinomycetes for the biotransformation of GAA in the deep 48-well microplate, four of the 283 studied strains (AI 034, AI 045, AI 074, and AI 125) were screened and might have similar biotransformation activity according to the ultra-performance liquid chromatography (UPLC) analyses. Therefore, one representative Streptomyces strain (AI 045) was selected for scaled-up fermentation to isolate the compound from the biotransformation. To make readers clear, we added the description in Materials and Methods at Line 98-101 of Page 3 in the revised manuscript.

6.        Line 56: Which is the composition of AI medium?

Author’s Reply:

        Thank you very much for your comment. We added the composition of AI medium at Line 74-76 of Page 2 in the revised manuscript.

7.        Line 60-61: Please could you insert some details about the isolation method.

Author’s Reply:

        Thank you very much for your comment. We added the details about the isolation method at Line 81-88 of Page 2 in the revised manuscript.

8.        Line 72: "The AI 045 strain was cultured in a 1000-mL baffled Erlenmeyer flask containing 100 mL of AI 72 medium and 20 mg/L of GAA"  Why did you use such a low substrate concentration?

Author’s Reply:

        Thank you very much for your comment. There are two reasons why we used 20 mg/L of GAA in our screening experiments: (1). The biotransformation activity was evaluated by sensitive UPLC method, which can detect as low as 1 mg/L of GAA or the produced metabolites. (2). The price of GAA is so high (50,000 NTD/g) that we only use low GAA concentration as long as detectable.

9.        Line 80: "spectroscopy (NOESY), and correlation spectroscopy (COSY) 83 spectra were recorded on a Bruker AV-700 NMR spectrometer (Bruker Corp., Billerica, MA, USA) at 84 ambient temperature. Standard pulse sequences and parameters were used for the NMR experiments, 85 and all chemical shifts were reported in parts per million (ppm, δ)." Which solvent did you use for NMR analysis?

Author’s Reply:

        Thank you very much for your comment. The solvent we used for NMR analysis was MeOH-d4. We added the information both in Line 108 of Page 3 in the main text and in the Figure legend of supplementary materials from Figure S1-S4 in the revised manuscript.

10.     Line 115-116: "From a 116 2.1-L fermentation broth containing 42 mg GAA, 2.9 mg of compound (1) was isolated."  This yield is very low and I think that this process should be improved to increase yields. It may be useful to monitor fermentation parameters such as pH, pO2 and temperature and to evaluate the initial substrate concentration.

Author’s Reply:

        Thank you very much for your comment. Biotransformation by using whole-cells as biocatalysts usually has lower efficiency than that by using purified enzyme, which could be produced through genetic engineering. Therefore, cloning of the putative genes encoding the catalytic enzymes in the biotransformation is a worthy strategy. The work is undergoing in our laboratory. After cloning the gene encoding the enzyme responsible for the biotransformation of GAA, the biotransformation could be conducted in a higher efficiency and much more amount of the new compound could be obtained.

11.     Line 132-133: "Therefore, the possible reactions to produce 3-O-acetyl GAA by the Streptomyces sp. AI045 strain might be also involved in two-stepwise reaction due to similar side-chain structures."  It would be very interesting to be able to verify this hypothesis, if it is not possible to recover the reaction intermediate, it would be useful to try to isolate the enzyme.

Author’s Reply:

        Thank you very much for your comment. Yes, it is indeed that identifying the catalytic enzymes involving in the biotransformation and then verifying the two-stepwise biotransformation hypothesis are very interesting. As our reply to question 10 above, the work is undergoing in our laboratory.

Reviewer 2 Report

The aim of reviewed manuscript was microbial biotransformation of ganoderic acid A from Ganoderma lucidum fungi into 3-Ο-Acetyl ganoderic acid A by Streptomyces sp. isolated from soil niches.

Reviewed manuscript includes fragments which are writing in good English language but on the other hand I can find many examples written in wrong English with grammar mistakes, a scientific less style of writing which very often makes the text hard to understand (line 30 constituents – compounds, line 31 To date – so far, all sentence from line 34, line 137 But the difficulty is we failed to......, and many others).

The authors write that further, specialystic analyzes should be carried out to confirm the described results. For this reason they should rather put forward hypothesis not conclusions that need scientific proofs in the future (line 142).

The authors presented the Introduction text in almost identical way in comparison to “New Triterpenoid from Novel Triterpenoid 15-O-Glycosylation on Ganoderic Acid A by Intestinal Bacteria of Zebrafish“, published in Molecules 2018. It is not well seen that two publications, even if written by the same Authors, contain such a very similar text.

I have no objections to the quality of provide results.

Below please find other suggestions:

1.     In Abstract, In my opinion 3-O-acetyl GAA should be called compound obtained from GAA or obtained after biotransformation of GAA, NOT „Biotransformed metabolite“, because it is a product of biotransformation not biotransformed compound.

2.     In Introduction section line 30, Authors mentioned about „some diseases“, what exactly diseases can be prevention by the G. lucidum? Examples of specific diseases should be listed despite giving the source of citation in brackets.

3.     In such a short communication Authors have mentioned three times about „300 different triterpenoids isolated from Ganoderma spp. In my opinion there is no need to repeat this information 3 times. If it is so important maybe Authors should list and briefly describe a few of the most important ones?

4.     Line 35, what exactly does mean „Ganoderma quality“? What kind of quality the authors had in mind?

5.     Line 37-40, the mechanism of influence of GAA on breast cancer cells should be described more clearly for readers.

6.     Line 41, I cant agree that „ searching new compounds is the basic step in the development of new drugs“. Researches on discover or create a new drugs is a extlemelly and many-steps experimental machiniery which involves many factors that’s why it should not be written that searching compounds is the basic step.

7.     Lines 43-44, „Precursor compounds were biotransformed by microorganisms or enzymes to form new 43 compounds. Additionally, some modifications on the functional groups of the precursors through the microbial biotransformations improve the bioactivities of the precursor compounds „these sentences are definitely too general and do not bring any specific information to this study, details, specific examples are required.

8.     Authors should highlight the novelty of Their research in the Introduction section and underline the scientific value of Their work.

9.     Materials and methods section, I believe that it is not necesary to repeat the same methods which were describe in previous publication. Hovewer actual version of manuscript is lack of detailes which could be helpfull for other scientific to replicate similar experiments. For example what excactly reagents and solvents have been used? What are „commercially available sources“?

10.  Authors should provide accurate data on the number of isolates tested and next chosen to further analysis in section Materials and methods in place of Results and Discussion

Author Response

To Reviewer 2:

The authors presented the Introduction text in almost identical way in comparison to “New Triterpenoid from Novel Triterpenoid 15-O-Glycosylation on Ganoderic Acid A by Intestinal Bacteria of Zebrafish“, published in Molecules 2018. It is not well seen that two publications, even if written by the same Authors, contain such a very similar text.

 Author’s Reply:

        Thank you very much for your comment. The Introduction was rewritten so that the sentences were different from those in our previous publication. In addition, the publication was also cited in the present study.

1.        In Abstract, In my opinion 3-O-acetyl GAA should be called compound obtained from GAA or obtained after biotransformation of GAA, NOT „Biotransformed metabolite“, because it is a product of biotransformation not biotransformed compound.

Author’s Reply:

Thank you very much for your reviewing and helpful comments. We changed the description “biotransformed metabolite’ to “the produced compound obtained from the biotransformation of GAA” at Line 22 of Page 1 in the revised manuscript.

2.        In Introduction section line 30, Authors mentioned about „some diseases“, what exactly diseases can be prevention by the G. lucidum? Examples of specific diseases should be listed despite giving the source of citation in brackets.

Author’s Reply:

        Thank you very much for your comment. We added the diseases what supplement of Ganoderma lucidum could prevent and changed the sentence to " Ganoderma lucidum (in Chinese, “Lingzhi”) has been used as a nutritional supplement in immunomodulatory and antitumor activities [1]." at Line 32-33 of Page 1 in the revised manuscript.

3.          In such a short communication Authors have mentioned three times about „300 different triterpenoids isolated from Ganoderma spp. In my opinion there is no need to repeat this information 3 times. If it is so important maybe Authors should list and briefly describe a few of the most important ones?

Author’s Reply:

        Thank you very much for your comment. We changed the sentence “more than 300 triterpenoids have been isolated from G. lucidum” to “many Ganoderma triterpenoids have been isolated and studied” at Line 61-62 of Page 2 and the description “more than 300 biosynthetic triterpenoids” to “numerous triterpenoids’ at Line 186 of Page 6 in the revised manuscript.

4.        Line 35, what exactly does mean „Ganoderma quality“? What kind of quality the authors had in mind?

Author’s Reply:

        Thank you very much for your comment. Ganoderma lucidum has been obtained from submerged culture. However, the quantity of Ganoderma triterpenoids obtained from the fermentation process varied dramatically based on fermentation parameters, such as pH and dissolved oxygen [4]. Due to a unique component in Ganoderma, GAA is used as a marker component for evaluating Ganoderma quality. To make readers clear, we added the above description at Line 36-41 of Page 1 in the revised manuscript.

5.        Line 37-40, the mechanism of influence of GAA on breast cancer cells should be described more clearly for readers.

Author’s Reply:

        Thank you very much for your comment. We added more detail about the inhibition of human breast cancer cells by GAA “Some bioactivities of GAA have been proven [5-10]. In particular, GAA was proven to play a role in antitumor activity in highly invasive human breast cancer cells [5], human liver cancer cells [6], and human hepatocellular carcinoma [7]. GAA was found to suppress the growth and invasiveness of human breast cancer cells by inhibiting transcription factor activator protein-1 (AP-1) and nuclear factor-kB (NF-kB), resulting in the down-regulation of cyclin-dependent kinase 4 (Cdk4) expression and the suppression of the secretion of urokinase-type plasminogen activator (uPA), respectively [5].” at Line 41-49 of Page 1-2 in the revised manuscript.

6.        Line 41, I cant agree that „ searching new compounds is the basic step in the development of new drugs“. Researches on discover or create a new drugs is a extlemelly and many-steps experimental machiniery which involves many factors that’s why it should not be written that searching compounds is the basic step.

Author’s Reply:

        Thank you very much for your comment. We changed the description “the basic step” to “an important step” at Line 52 of Page 2 in the revised manuscript.

7.        Lines 43-44, „Precursor compounds were biotransformed by microorganisms or enzymes to form new 43 compounds. Additionally, some modifications on the functional groups of the precursors through the microbial biotransformations improve the bioactivities of the precursor compounds „these sentences are definitely too general and do not bring any specific information to this study, details, specific examples are required.

Author’s Reply:

        Thank you very much for your comment. We added an example “For example, soy isoflavone daidzein was hydroxylated by Aspergillus oryzae to form 6-hyroxydaidzein or 8-hydroxydaidzein. These two compounds have not been identified in nature. Moreover, the two compounds possessed over 10-fold more anti-tyrosinase activity than their precursor daidzein [15].” in the description at Line 57-60 of Page 2 in the revised manuscript.

8.        Authors should highlight the novelty of Their research in the Introduction section and underline the scientific value of Their work.

Author’s Reply:

        Thank you very much for your comment. The novelty and the scientific value of the present work is “Many Ganoderma triterpenoids have been isolated and studied; however, few studies have been published on the microbial biotransformation of Ganoderma triterpenoids. The present study investigated biotransformation of Ganoderma triterpenoid GAA and identified one soil-isolated Streptomyces sp. AI 045, making it the first case in which a microbe was found to biotransform triterpenoid to its 3-O-acetyl derivative. This is the first time that the acetylation metabolite 3-O-acetyl GAA has been identified.”, the above descriptions were at Line 61-63 of Page 2 in Introduction of the revised manuscript and at Line 188-192 of Page 6 in Conclusion of the revised manuscript.

9.          Materials and methods section, I believe that it is not necesary to repeat the same methods which were describe in previous publication. Hovewer actual version of manuscript is lack of detailes which could be helpfull for other scientific to replicate similar experiments. For example what excactly reagents and solvents have been used? What are „commercially available sources“?

Author’s Reply:

        Thank you very much for your comment. We added more detail descriptions about experimental process in the Materials and Methods of the revised manuscript, including: the composition of AI medium at Line 74-76 of Page 2 in the revised manuscript and the details isolation method at Line 81-88 of Page 2 in the revised manuscript.

10.     Authors should provide accurate data on the number of isolates tested and next chosen to further analysis in section Materials and methods in place of Results and Discussion.

Author’s Reply:

        Thank you very much for your comment. During the screening of soil actinomycetes for the biotransformation of GAA in the deep 48-well microplate, four of the 283 studied strains (AI 034, AI 045, AI 074, and AI 125) were screened and might have similar biotransformation activity according to the ultra-performance liquid chromatography (UPLC) analyses. Therefore, one representative Streptomyces strain (AI 045) was selected for scaled-up fermentation to isolate the compound from the biotransformation. To make readers clear, we added the description in Materials and Methods at Line 98-101 of Page 3 in the revised manuscript.

Reviewer 3 Report

The authors successfully isolated the Streptomyces sp. AI045 in the soil, which had the ability to biotransform ganoderic acid A (GAA) to 3-O-acetyle GAA. Moreover, 3-O-acetyle GAA was identified for the first time, although its function as a drug was still unknown. The strategy for the research and analytical method used in the present study were reasonable. Existing literature is properly cited, and the manuscript is clear, concise, and well organized. It might be valuable information for the field of research of microbial biotransformation, and the scientific impact and novelty of this article are suitable for the publication.

Author Response

To Reviewer 3:

The authors successfully isolated the Streptomyces sp. AI045 in the soil, which had the ability to biotransform ganoderic acid A (GAA) to 3-O-acetyle GAA. Moreover, 3-O-acetyle GAA was identified for the first time, although its function as a drug was still unknown. The strategy for the research and analytical method used in the present study were reasonable. Existing literature is properly cited, and the manuscript is clear, concise, and well organized. It might be valuable information for the field of research of microbial biotransformation, and the scientific impact and novelty of this article are suitable for the publication.

Author’s Reply:

Thank you very much for your reviewing and your support for the publication.

Reviewer 4 Report

Dear authors,

This is a useful study since these chemicals can act against breast cancer cells. Kudos for investigating such global problem. I have no significant objections, only small question: is supplementary material in Catalyst template?

Author Response

To Reviewer 4:

This is a useful study since these chemicals can act against breast cancer cells. Kudos for investigating such global problem. I have no significant objections, only small question: is supplementary material in Catalyst template?

Author’s Reply:

Thank you very much for your reviewing and the comment. We have deleted the Catalyst template in the revised supplementary material.